# Psychometric Properties of the Italian Version of the 25-Item Hikikomori Questionnaire for Adolescents

**DOI:** 10.3390/ijerph191610408

**Published:** 2022-08-21

**Authors:** Simone Amendola, Fabio Presaghi, Alan Robert Teo, Rita Cerutti

**Affiliations:** 1Department of Dynamic and Clinical Psychology and Health Studies, Sapienza University of Rome, 00185 Rome, Italy; 2Department of Psychology of Development and Socialization Processes, Sapienza University of Rome, 00185 Rome, Italy; 3Center to Improve Veteran Involvement in Care (CIVIC), VA Portland Health Care System, Portland, OR 97239, USA; 4Department of Psychiatry, Oregon Health & Science University, Portland, OR 97239, USA

**Keywords:** social isolation, social withdrawal, social avoidance, youth, young people

## Abstract

Hikikomori is a form of social withdrawal that is commonly described as having an onset during adolescence, a life stage when other psychiatric problems can also emerge. This study aimed to adapt the 25-item Hikikomori Questionnaire (HQ-25) for the Italian adolescent population, examining its psychometric properties; associations between hikikomori and psychoticism, depression, anxiety, problematic internet use (PIU), psychotic-like experiences (PLEs), to confirm convergent validity of the HQ-25; and the interaction effect between symptoms of hikikomori and PIU in predicting PLEs. Two-hundred and twenty-one adolescents participated in the study. Measures included the HQ-25, the Psychoticism subscale of the Personality Inventory for the Diagnostic and Statistical Manual of Mental Disorders, the Depression and Anxiety subscales of the Brief Symptom Inventory, the Internet Disorder Scale, and the Brief Prodromal Questionnaire. Data showed a satisfactory fit for a three-factor model for the HQ-25 that is consistent with the original study on the HQ-25. Three factors (socialization, isolation, and emotional support) were associated with psychopathology measures. Six participants reported lifetime history of hikikomori. Symptoms of hikikomori and PIU did not interact in predicting PLEs. This is the first study to validate the HQ-25 in a population of adolescents. Findings provide initial evidence of the adequate psychometric properties of the Italian version of the HQ-25 for adolescents.

## 1. Introduction

*Hikikomori* refers to a serious form of social withdrawal in which the individual voluntarily and prolongedly withdraws from social relationships isolating himself/herself at home. Marked social isolation at home for at least 6 months associated with significant functional impairment or distress are suggestive of hikikomori [1,2]. The term hikikomori, which originated in Japan, can refer both to the individual who is affected by it and to the condition itself. The phenomenon was initially described within Japan, and some authors have argued that hikikomori fulfilled the criteria of what was then called “culture-bound syndromes” (now called cultural syndromes), suggesting additional research into whether it represented a novel form of psychopathology more globally [3]. Umeda and Kawakami [4] analyzed data derived from the World Mental Health Japan survey collected from 2002 to 2006. Lifetime hikikomori was reported by 1.2% of people aged 20 to 49 years. Importantly, the authors showed that, in over 50% of cases, onset was during adolescence. The Cabinet Office of Japan [5] published the results of an epidemiological survey on hikikomori among people aged 15 to 39 years, confirming previously observed prevalence and age of onset: 12.2% of individuals with hikikomori reported that their current situation started before they were 14 years old, while 30.6% responded that the withdrawn behavior started between 15 and 19 years old. Thus, hikikomori onset usually occurs during adolescence and young adulthood, at least in Japan. The duration of social withdrawal is generally long and severe, although variable [6]. In the study of Umeda and Kawakami [4], 49% of people afflicted by hikikomori reported a duration of 6–11 months. On the contrary, Tajan et al. [5] showed that only 12% of people with hikikomori socially withdrew for less than a year, while over 75% of cases were socially isolated at home for more than 3 years. Thus, hikikomori represents a source of concern for societies and their economic and public health systems [7].

In the last decade, cases of hikikomori have been reported worldwide both in adults [6,8,9,10,11,12,13,14,15] and adolescents [16,17,18]. In Italy, Ranieri [19,20] described three cases of adolescent girls with hikikomori. They showed typical symptoms of hikikomori such as prolonged social withdrawal at home and no social relationships. During interventions such as individual psychotherapy and/or home-visiting, substantial improvements were observed, although the adolescents were not able to resume attending school [19,20]. Further, more recent studies [21,22] describe two cases of adolescents with hikikomori and comorbid disorders such as videogame addiction and narcolepsy.

Empirical studies on hikikomori have also been published in non-Asian countries [23,24,25,26,27]. Our previous findings with samples of young adults showed that lifetime episodes of hikikomori were reported by 15.8% and 4.3% of participants with and without comorbid psychopathology, respectively [28], whereas the prevalence of lifetime hikikomori was 1.1% in a sample of adults aged 18–50 years [29]. Findings of a different study [23] indicated that 20.9% and 2.7% of university students from Nigeria and the United States, respectively, reported lifetime hikikomori.

To date, few empirical studies [30,31] have specifically focused on adolescents, although clinical cases of hikikomori have been described in adolescence and the first onset of hikikomori is in the majority of cases during this life phase. Moreover, school refusal and a lack of social relationships and social interaction during the sensitive time of adolescence may increase subsequent interpersonal difficulties and symptoms of psychopathology, such as depression, anxiety, and loneliness [31,32,33]. Lee et al. [31] and Hamasaki et al. [30] showed that adolescents with hikikomori reported more symptoms of psychopathology than those without hikikomori in South Korea and Japan, respectively. Moreover, there has been considerable interest in the association between personality dysfunction and hikikomori in adulthood [26,27,28,34]. According to the Alternative Model of Personality Disorder, psychoticism refers to a disconnection from reality, a tendency for illogical thought patterns, and unusual behaviors [35,36]. It has been postulated that it reflects an evolved psychological system concerning a reality model for action with the function of creating mental models of the external environment for planning behaviors [37]. Previous research showed associations between psychoticism and negative emotions, self-esteem, loneliness and motivational aspirations (i.e., wealth, fame, image) [38].

### 1.1. Symptoms of Hikikomori, Psychotic-like Experiences and Problematic Internet Use

Although hikikomori is generally considered to be different from a psychotic condition mainly due to the absence of the positive or negative symptoms of schizophrenia, it is possible that it may include schizophrenia before a definitive diagnosis [1], considering that withdrawal is one of the primary symptoms of psychosis [39,40,41]. On the one hand, it is possible that psychosis leads to hikikomori as a symptom. However, on the other hand, hikikomori could be the cause of coexistent psychosis and, at this time, either trajectories cannot be ruled out [1].

Healthy individuals may develop psychotic-like experiences (PLEs) as a response to physical isolation and sensory deprivation [42,43]. PLEs such as perceptual disturbances, unusual thinking, suspiciousness, grandiosity, and disorganized communication are prevalent among the general population [44,45], especially among adolescents [46,47]. According to the psychosis proneness persistence–impairment model, PLEs reflect the existence of a “psychosis severity continuum” and are not necessarily associated with the presence of a disorder [48]. However, PLEs are considered markers for a wide range of psychiatric symptoms [49,50] and may help in identifying adolescents and emerging adults at high clinical risk for psychosis [51]. Recently, an association between lifetime psychotic experiences, especially delusional, and lifetime hikikomori has been demonstrated in a sample of adult community residents in Japan [52].

Moreover, researchers and clinicians have discussed the possible link between hikikomori and the spread of the internet and new technologies use (e.g., computers, smartphones, videogame consoles) [1,53]. Associations between hikikomori and problematic internet use (PIU) have been demonstrated both in adolescents [30,31] and adults [26,29,54] with some exception [28]. At the same time, empirical studies have shown significant associations between PIU and PLEs in samples of young people [55,56,57,58,59], and the possible intersection between the diagnoses of hikikomori, PIU, and PLEs has recently been discussed [60]. Importantly, reports of clinical cases [61,62,63,64,65,66,67,68,69] have suggested that the excessive use of new technologies and PIU, when associated with social withdrawal, may constitute an important risk factor that is able to unmask psychotic vulnerability or foster the emergence of psychotic symptoms. The social deafferentation hypothesis of psychosis posits social isolation as a stressor which may trigger the development of psychotic experiences among individuals with pre-existing vulnerabilities [70]. A recent meta-analysis confirmed that there is a robust association between loneliness and psychotic symptoms in both clinical and non-clinical samples [71]. Several factors associated with characteristics of PIU can exert an effect on psychological distress and exacerbate the impact of social withdrawal on PLEs, such as withdrawal symptoms, use of the Internet as a substitute for reality, distortion of spatial perception and geographical distance, reduced involvement of the body, division between language, face movement, and non-verbal behavior, fluid notion of “truth”, transparency between private and public contents, malnutrition, and irregular sleep [65,66,69]. Thus, PIU could moderate the association between social withdrawal and PLEs, i.e., at high levels of PIU could correspond to a stronger association between social withdrawal and PLEs than at low levels of PIU.

### 1.2. The 25-Item Hikikomori Questionnaire (HQ-25) and Objectives of the Present Study

To our knowledge, no instrument has been published in English or Italian that is tailored to measure the symptoms of hikikomori in adolescence. Teo et al. [72] developed a self-report scale for the assessment of the severity of hikikomori symptoms, the 25-item Hikikomori Questionnaire (HQ-25), among a Japanese sample of 399 psychiatric and community setting participants aged 15 to 50 years. The authors, experts in clinical psychology, psychiatry, and hikikomori, first generated a list of 59 items according to the hikikomori literature. Their analyses pointed out robust psychometric properties and diagnostic accuracy for a 25-item questionnaire, and highlighted a three-factor solution, representing socialization, isolation, and emotional support. Items that load on the socialization factor describe problems/difficulties in social interactions and a preference for being alone rather than with others. The items of the Isolation factor mainly refer to withdrawal behaviors and physical isolation. Finally, the items of the emotional support factor explore the presence/absence of people providing the emotional aspects of social support. The HQ-25 demonstrated convergent validity with measures of loneliness and a preference for solitude and divergent validity through a negative correlation with a measure of social support [72]. Recently, the questionnaire has been used in two studies [28,54], and the Italian version of the HQ-25 [29] has been validated with a sample of Italian adults demonstrating adequate psychometric properties. Overall, the findings of these initial studies [29,54] showed no sex difference on the total HQ-25 score as well as on the three-factor scores. Further, they demonstrated that participants in a condition at risk of developing hikikomori reported higher symptoms of psychopathology than those not at risk of hikikomori. In light of the above findings and considering the increasing public health and scientific concerns around hikikomori in Italian adolescents [20,21,22,73,74,75,76,77,78], the first aim of the present study was to adapt the Italian version of the HQ-25 for adolescents and analyse its psychometric properties. The second aim was to explore the relationships between symptoms of hikikomori and the psychoticism personality domain of personality, symptoms of depression, anxiety, PIU, PLEs, and self-reported history of hikikomori to confirm the convergent validity of the HQ-25. The third aim was to test the interaction effect between symptoms of hikikomori and PIU in predicting PLEs (i.e., does the association between symptoms of hikikomori and PLEs vary according to PIU?) controlling for the effect of important covariates (i.e., psychoticism, depression, and anxiety).

## 2. Materials and Methods

### 2.1. Participants

This cross-sectional study included a convenience sample of 227 adolescents attending two secondary schools in central Italy (i.e., in the municipality of Rome) in January 2020. While we are unsure of the precise number of students at these schools who were eligible for this study, nearly all students at one of the schools, and approximately one-quarter at the other school, participated in the study. Data from 6 participants were excluded because at least one questionnaire was not completed. Thus, the final sample included 221 students (55.7% males, *n* = 123) aged 11–19 years (Mage = 15.15, SD = 2.91). Exclusion criteria for participation in the study were poor comprehension of written Italian and a history of significant neurological illness or brain injury.

### 2.2. Procedures

This study was conducted in order to examine the relationship between prolonged social withdrawal, problematic use of technologies, and mental health among Italian students. The aims of the study were illustrated to the headmasters and teachers of each school. Informed consent was obtained from both participants and their parents before enrolment in the study. After providing informed consent, questionnaires were briefly presented to the classrooms during school time. The administration lasted approximately 45 min. Participants filled out the questionnaires collectively in the classroom after brief instructions were provided. Anonymity of participants was ensured by assigning each participant a unique code. This study was approved by the Ethics Committee of the Department of Dynamic and Clinical Psychology and Health Studies, Sapienza University of Rome. The protocol for the research project conformed with the provisions of the Declaration of Helsinki.

### 2.3. Measures

#### 2.3.1. Prolonged Social Withdrawal

The *25-item Hikikomori Questionnaire* (HQ-25) [29,72] is a self-report questionnaire that evaluates the severity of symptoms of prolonged social withdrawal over the preceding 6 months. Typical psychological features and behavioral patterns of hikikomori, such as socialization (e.g., item 1 = “I stay away from other people”, item 23 = “I do not enjoy social interactions”), isolation (e.g., item 2 = “I spend most of my time at home”, item 24 = “I spend very little time interacting with other people”), emotional support (e.g., item 3 = “There really is not anyone with whom I can discuss matters of importance”, item 21 = “I have someone I can trust with my problems”), and a sense of alienation from society, are investigated. Participants respond on a 5-point Likert scale (from 0 = “strongly disagree” to 4 = “strongly agree”). The HQ-25 has a score range of 0–100. Higher values indicate higher symptomatology. The authors demonstrated that a cutoff score of 42 was able to discriminate between individuals at risk for hikikomori and those not at risk [72].

The Italian version of the HQ-25 showed good psychometric properties in a sample of Italian adults aged 18–50 years [29]. However, to verify that the adolescents adequately understood the HQ-25 items, five adolescent students from different classes piloted the questionnaire, reporting issues with respect to understanding or interpretating items. Five of the 25 items (i.e., items 13, 15, 16, 23, 24) were revised in order to simplify the wording so as to promote their comprehension by younger adolescents. The Italian version of the HQ-25 for adolescents is presented in Appendix B.

#### 2.3.2. Self-Reported History of Hikikomori

The presence of lifetime episodes of hikikomori was examined in the majority of participants (78.7%, *n* = 174) because this questionnaire was added after data collection from participants had begun. This study used the definition of hikikomori proposed by Kato et al. [1,2]. The following symptoms were investigated through a questionnaire designed for and used in two previous studies [28,29]: (1) marked social isolation in one’s home (i.e., leaving the house 3 days/week or less) and (2) duration of social withdrawal of at least six months. Moreover, we examined the characteristics of students with a duration of social withdrawal longer than three months but less than six (i.e., *pre-hikikomori*).

#### 2.3.3. Personality

*Psychoticism subscale*—*Personality Inventory for DSM-5 Brief Form* (PID-5BF) [35,79]. Five items examine the psychoticism domain with four response choices distributed on a 4-point Likert scale (from 0 = “very false/often false” to 3 = “very true/often true”). The maladaptive functioning of the individual linked to the psychoticism domain of personality increases as the score increases. In the present study, Cronbach’s alpha was 0.74.

#### 2.3.4. Psychopathology

The *Internet Disorder Scale* (IDS-15) Italian version [80,81] is a self-report scale composed of 15 items exploring the severity and the impact of problematic internet use. Items focus upon users’ online leisure activity from any device with internet access over the past year. Respondents rated each item on a 5-point Likert scale (from 1 = “strongly disagree” to 5 = “strongly agree”). The total score range is 15 to 75, with higher scores indicating higher degrees of problematic internet use. In the present study, Cronbach’s alpha of the scale was 0.83.

The *Brief Symptom Inventory* (BSI) [82] is a 53-item self-report instrument that assesses nine primary psychological symptom dimensions during the past seven days. The respondents rank each item on a 5-point Likert scale (from 0 = “not at all” to 4 = “extremely”). Higher scores indicate a higher severity of psychological distress. For the purposes of the present study, depression and anxiety dimensions were used. The Italian version of the BSI has acceptable psychometric properties [83]. In the present study, Cronbach’s alpha of the two dimensions was 0.84 (depression) and 0.81 (anxiety).

The *Brief Prodromal Questionnaire* (PQ-B) [44,84,85,86] is a self-report questionnaire including 21 items with a dichotomous response (yes versus no) used to screen individuals for the positive symptoms of psychosis experienced over the past month. For each endorsed symptom, respondents rate whether they found it distressing or impairing (from 1 = “strongly disagree” to 5 = “strongly agree”). The PQ-B has been adopted as a screening tool using the total number of items endorsed (“symptom total score”), the number of items that are identified as distressing (“distressing item total score”) (both range from 0 to 21), and the “total distress score” (range from 0 to 105), with the latter method being generally recommended [87]. For the purposes of our study, we used the total distress score. Nonetheless, we reported the results of the symptom total score and specific psychotic symptoms according to sex and risk for hikikomori as Appendix A. The Italian version of the PQ-B showed good psychometric properties [44,85,86]. In the present study, Cronbach’s alpha was 0.79.

### 2.4. Statistical Analysis

First, we checked the normality of distribution for all 25 items of the HQ-25 by inspecting whether skewness and kurtosis parameters were within the bounds of ± 2.00. Results showed that at least 4 items reported a skewness and/or a kurtosis exceeding those bounds. To confirm the 3-factor structure of the HQ-25, we performed a confirmatory factor analysis. Given that not all items were normally distributed, we used the Unweighted Least Squares (ULS) estimator with robust standard error estimates (ULSM). As measures of model fit, we considered a non-significant robust Chi squared statistic, in addition to the following fit indices: the Comparative Fit Index (CFI) and the Bentler–Bonnett Non-Normed Fit Index (NNFI) (both above 0.90), the Root Mean Square Error of Approximation (RMSEA) and its 90% Confidence Interval (C.I.) (below 0.08), and the Standardized Root Mean Square Residual (SRMR) (below 0.08).

We also estimated latent correlations among factors to investigate the relationships between the HQ-25’s factors and the psychological constructs of interest (i.e., psychoticism domain of personality, problematic internet use, symptoms of depression and anxiety and distress due to psychotic-like experiences) to confirm the convergent validity of the HQ-25. Specifically, the measurement models of all questionnaires/subscales were entered simultaneously in a correlated factor model to estimate the latent correlation between all constructs. The advantage of modeling a latent correlation matrix compared to a matrix consisting of correlations between the total scale scores is that total score correlations are typically attenuated due to the measurement error in the questionnaire item scores, while latent correlations are not affected by this problem, and they therefore paint a less biased picture of the associations between constructs [88].

Reliability was computed both with Cronbach alpha as well as with McDonald’s Omega [89]. For estimating significant differences between groups (i.e., according to sex and HQ-25 cutoff), we used Student-t statistics (and Cohen’s *d* as an estimate effect size) or Chi-square test of independence. Furthermore, HQ-25 total and subscales scores were compared according to self-reported history of hikikomori.

Finally, we were interested in verifying whether the relationship between symptoms of hikikomori and distress due to PLEs varied depending on PIU. For this, we tested a structural equation model including the interaction between PIU and symptoms of hikikomori as evaluated using the HQ-25.

All statistical analysis were performed using R [90] with the lavaan package [91], the psych package [92], and the coefficientalpha package [93].

## 3. Results

Confirmatory factor analysis showed a satisfactory fit for the three-factor model (Robust χ^2^(272) = 547.283, *p* < 0.001, Robust CFI = 0.96, Robust NNFI = 0.96, Robust RMSEA = 0.068, 90% C.I. for Robust RMSEA: 0.060; 0.077; SRMR = 0.067). The one-factor model (the baseline comparison model) also reported a good fit to the data (Robust χ^2^(275) = 599.66, *p* < 0.001, Robust CFI = 0.96, Robust NNFI = 0.95, Robust RMSEA = 0.073, 90% C.I. for Robust RMSEA: 0.065; 0.081; SRMR = 0.071). However, the fit of the three-factor model was significantly better than that of the one-factor model (Δχ^2^(3) = 36.03, *p* < 0.001).

Completely standardized factor loadings for the three-factor structure as well as latent correlations among the three factors are shown in Table 1. Socialization had a strong positive correlation with both isolation (phi = 0.908, *p* < 0.001) and emotional support (phi = 0.743, *p* < 0.001), as did isolation with emotional support (phi = 0.691, *p* < 0.001).

The Cronbach alpha and Omega values demonstrated satisfactory reliability for the subscales socialization (Cronbach alpha = 0.85; 95% C.I. Cronbach alpha: 0.81 to 0.89; Omega = 0.86, 95% C.I. Omega: 0.81 to 0.90) and isolation (Cronbach alpha = 0.75; 95% C.I. Cronbach alpha: 0.67 to 0.82; Omega = 0.76, 95% C.I. Omega: 0.69 to 0.83), whereas reliability of the subscale emotional support (Cronbach alpha = 0.63; 95% C.I. Cronbach alpha: 0.54 to 0.72; Omega = 0.64, 95% C.I. Omega: 0.56 to 0.73) was questionable.

### 3.1. Relationship between HQ-25 Factors and Other Psychological Constructs

To investigate the relationships between scores on the three HQ-25s’ factors and scores on the questionnaires/subscales of interest, we computed the latent correlations (Robust χ^2^(1052) = 1796.588, *p* < 0.001, Robust CFI = 0.97, Robust NNFI = 0.97, Robust RMSEA = 0.053, 90% C.I. for Robust RMSEA: 0.049; 0.057; SRMR = 0.066). All three HQ-25s’ factors positively correlated with psychoticism, PIU, depression, anxiety, and total distress due to PLEs, except for the relationship between emotional support and PIU which was non-significant (*p* = 0.102) (Table 2). See also Appendix A. As requested during the review process, we conducted a series of tests of invariance to investigate whether the three HQ factors were equally associated with PLEs’ total distress and whether socialization and isolation were equally associated with psychoticism, PIU, depression and anxiety. To test the first hypothesis, we constrained the latent correlations of the three HQ factors with PLEs’ total distress to be equal. Results from the likelihood ratio test between unconstrained and constrained models showed that the fit of the constrained model was not significantly different from the fit of the non-constrained model (χ^2^(2) = 2.34, *p* = 0.3099). For testing the second hypothesis, we constrained latent correlations of socialization and isolation with psychoticism to be equal. The same constraints were applied to the latent correlations with PIU, depression, and anxiety. Results from the likelihood ratio test showed that the fit of this second constrained model was not statistical different from the constrained model used for the first hypothesis (χ^2^(4) = 4.13, *p* = 0.3883). In conclusion, the two hypotheses were supported by the results.

### 3.2. Mean Differences According to Sex

As shown in Table 3, we found significant differences between girls (*n* = 98) and boys (*n* = 123) for the socialization subscale, Internet Disorder Scale, and depression and anxiety dimensions of the BSI. Girls reported higher scores than boys on those questionnaires/subscales. Appendix A show the prevalence of PLEs according to sex and the results of the Chi-squared tests.

### 3.3. Differences between Participants According to HQ-25 Cutoff

Considering the proposed cutoff score of 42 or above on the HQ-25 as indicating being “at-risk of hikikomori”, we found that 12.2% (*n* = 27) of the sample exceeded the threshold score. Sex was not associated with a risk for hikikomori (among females: 14.3%, *n* = 14; among males: 10.6%, *n* = 13). The “at-risk” group (M = 15.93, SD = 2.91) reported a non-significant (*t*(219) = −1.55, *p* = 0.12, Cohen’s *d* = 0.338) higher mean age than the “not at-risk” group (M = 15.05, SD = 2.91).

We further investigated whether the group “at-risk” for hikikomori reported significantly higher scores than the group “not at-risk” for hikikomori on the questionnaires/subscales that participants filled in along with the HQ-25 (Table 4). Results showed that the group “at risk” for hikikomori reported significantly higher scores of psychoticism, PIU, depression, anxiety, and total distress associated with PLEs than the “not at-risk” group. Finally, Appendix A reports significant differences in PLEs according to the risk for hikikomori.

### 3.4. Scores on the HQ-25 According to Lifetime Episodes of Hikikomori and Pre-Hikikomori

Table 5 shows the HQ-25 total score and subscores for lifetime hikikomori and pre-hikikomori compared with participants who did not report lifetime episodes of prolonged social withdrawal. Six participants (3.4% of the subsample; among females: 4.3%, *n* = 3; among males: 2.9%, *n* = 3) reported lifetime episodes of hikikomori (*n* = 3, boys *n* = 2) or pre-hikikomori (*n* = 3, boys *n* = 1). The mean age of participants who reported lifetime episodes of hikikomori and pre-hikikomori (M = 16.3, SD = 2.3) seemed to be higher than that of the participants who did not report episodes of social withdrawal (M = 14.39, SD = 2.80).

### 3.5. Moderation Effect of PIU on the Relationships between Symptoms of Hikikomori and PLEs

Finally, we ran a structural equation model (Robust χ^2^(412) = 648.263, *p* < 0.001, Robust CFI = 0.88, Robust NNFI = 0.87, Robust RMSEA = 0.055, 90% C.I. for Robust RMSEA: 0.047; 0.063; SRMR = 0.070) to explore the relationships between age, sex, psychoticism, depression, anxiety, symptoms of hikikomori, PIU, and distress due to PLEs, as well as to explore the role of PIU as a moderator of the association between symptoms of hikikomori and distress due to PLEs. As shown in Table 6, psychoticism and anxiety were positively associated with total distress due to PLEs. We did not find evidence on the role of PIU as a moderator of the relationship between symptoms of hikikomori and total distress due to PLEs.

## 4. Discussion

Overall, our findings demonstrated the initial reliability and validity of the Italian version of the HQ-25 for adolescents, supporting its use with this population to help with the screening for hikikomori. We adapted the HQ-25 by simplifying the wording of five of the 25 items. The CFA confirmed the three-factor structure of the instrument consisting of socialization, isolation and emotional support. Further, we analyzed the convergent validity of the HQ-25 factors by correlating their scores with scores measuring psychoticism, depression, anxiety, PIU and PLEs. Latent correlations in the expected direction were shown to support the validity of the instrument in its current slightly revised form for Italian adolescents.

Another important aspect of this study was our test of the hypothesis that PIU may represent a risk factor that moderates the relationship between symptoms of social withdrawal and distress due to PLEs. Several reports [61,62,63,64,65,66,67,68,69] of clinical cases seem to support the explanation in which severe social withdrawal and PIU mutually reinforce each other over time, with a progressive abandonment of social and school or work activities until symptoms of psychosis emerge. However, our findings showed that the association between symptoms of hikikomori and total distress due to PLEs did not vary according to PIU, after controlling for the effects of individual vulnerability factors, distress due to PLEs was primarily associated with the psychoticism trait domain of personality and symptoms of anxiety. Despite this, it is possible that the complex association between hikikomori, PIU, and psychotic symptoms became evident during young adulthood and adulthood, in accordance to the reports of clinical cases [61,62,63,64,65,66,67,68,69] and previous studies [52,56]. Thus, longitudinal studies with representative samples of different population are needed.

Some of our findings provide confirmation of early data on the HQ-25 in a previously unexamined study population, which expands our understanding of the generalizability of the HQ-25. For instance, the mean score of 24.66 (SD = 15.03) on the HQ-25 was in line with the findings of previous studies using the HQ-25 with non-clinical populations [28,29,54]. Similar to other studies [29,54], no sex difference was detected for the total score as well as for the isolation and emotional support factors. More than twelve percent of the sample exceeded the threshold score of the HQ-25, indicating an at-risk condition for hikikomori according to the cutoff score as reported in the original study [72]. This prevalence is consistent with those of previous studies conducted in Italy [28,29] and lower than that observed in a sample of Japanese college and university students [54].

Significant and positive associations emerged between symptoms of hikikomori, risk for hikikomori, and symptoms of psychopathology such as psychoticism, depression, anxiety, PIU, and PLEs. These results are in line with those of previous studies [26,27,30,31,34,52,54] and justify the increasing attention on hikikomori as the results of a dysfunctional process of adaptation to increased social and environmental demands [28,29,74]. Moreover, we showed that participants at risk for hikikomori were more likely to report psychotic symptoms such as unusual or bizarre beliefs, worry that something is wrong with the mind, feelings of non-existence, that the world does not exist, not-being in control of ideas and thoughts, and feeling of being watched by other people, than participants not at risk for hikikomori (see Appendix A). Accordingly, the results of a recent study [52] highlighted that the association between hikikomori and delusional experiences persisted even after controlling for sociodemographic characteristics and mental disorders during the last year, whereas the same did not happen when considering the relationship between hikikomori and hallucinatory experiences. These results are consistent with research findings suggesting that different indicators of social isolation may contribute to the maintenance of delusions (especially persecutory beliefs) by limiting involvement in social opportunities that could help individuals in reviewing and/or disconfirming their beliefs [94,95,96]. Thus, in taking care of adolescents with elevated symptoms of hikikomori, clinicians need to evaluate and, possibly, include in the treatment processes other important factors such as PIU and PLEs. Additionally, the specific relationship between symptoms of hikikomori and individual psychotic psychopathologies should be considered.

In the present study, six adolescent students reported a lifetime episode of hikikomori (1.7%, *n* = 3) and pre-hikikomori (1.7%, *n* = 3). In 2018, the Regional Education Office of the region Emilia-Romagna (Italy) carried out a survey involving 687 educational institutions out of a regional total of 702 primary and secondary schools [97]. The institutions reported that 99 children and adolescents who dropped out of school rarely left their home, 39 left their home only with their parents, and 39 never went out. Taken together, these initial results should stimulate an in-depth psychological reflection and investigation of the mental suffering of young people who are socially withdrawn [74]. Further research might investigate the use of the HQ-25 as a useful screening measure for identifying and directing adolescents particularly at risk for hikikomori to the appropriate mental health services, if needed.

Finally, girls showed a higher mean score for the socialization factor than boys indicating higher avoidance of social situations among girls. This difference could be related, in part, to the increased developmental challenges for girls [98] as well as body dissatisfaction, shyness, behavioural inhibition and internalizing of symptoms during adolescence [99,100,101,102]. Further research is needed to determine whether the difference observed indicates a real difference in socialization as measured using the HQ-25 or whether it is related to response style on the self-report.

This study has some limitations that need to be recognised. First, our findings cannot be generalized to other groups or populations such as adults or adolescents with psychiatric disorders, given that only school students participated in the present study. Further, the sample was a convenience sample from two schools. Due to this, the prevalence estimate should not be considered as a reliable or generalizable estimate of hikikomori. Second, data were collected using self-report measures and, thus, are susceptible to response biases. Third, we cannot exclude the possibility that only those students who were more interested in the topic may have been more prone to participate. Fourth, students who were in hikikomori at the time probably did not participate in the study. Moreover, we analysed psychometric properties as well as convergent validity of the questionnaire rather than criterion validity. Therefore, this limits the results of the study, despite the fact that the sample recruited was appropriate for the analysis carried out. Fifth, the cross-sectional design of the study does not provide an opportunity to demonstrate the temporal link between symptoms of hikikomori and the variables of interest. Therefore, predictive conclusions cannot be made. Sixth, the cutoff score used in the current study was not specific to an adolescent sample. Despite the limitations, the present study represents a relevant contribution to the scientific literature, bearing in mind the limited measures of hikikomori symptoms in adolescence. The HQ-25 seems to be an adequate screening tool for the investigation of symptoms of hikikomori. Additional work to determine an optimal cutoff to improve clinical validity in Italian and other samples of participants is warranted.

## 5. Conclusions

The present study provides initial evidence on the use of the HQ-25 as a sound self-report questionnaire to explore symptoms of hikikomori in Italian adolescents. The findings of this study expand the scientific literature on the associations between symptoms of hikikomori and psychopathology during adolescence. The use of the HQ-25 may help psychologists and other health professionals in evaluating symptoms of hikikomori in adolescents who avoid social situations and/or present numerous school absences.

## Figures and Tables

**Table 1 ijerph-19-10408-t001:** Completely standardized factor loadings among and latent correlations between the three factors of the HQ-25.

	Socialization	Isolation	Emotional Support
Item 1	0.721		
Item 4	0.328		
Item 6	0.519		
Item 8	0.634		
Item 11	0.558		
Item 13	0.690		
Item 15	0.527		
Item 18	0.673		
Item 20	0.732		
Item 23	0.526		
Item 25	0.531		
Item 2		0.479	
Item 5		0.575	
Item 9		0.715	
Item 12		0.466	
Item 16		0.308	
Item 19		0.690	
Item 22		0.460	
Item 24		0.621	
Item 3			0.648
Item 7			0.292
Item 10			0.547
Item 14			0.333
Item 17			0.446
Item 21			0.573
Socialization	1		
Isolation	0.908	1	
Emotional Support	0.743	0.691	1

Note. HQ-25: 25-item Hikikomori Questionnaire. All parameters are significant at *p* < 0.001 (except for Item 7, which is significant at *p* = 0.001). N = 221.

**Table 2 ijerph-19-10408-t002:** Latent correlations between the three HQ-25 factors and psychological constructs.

	Socialization	Isolation	Emotional Support
Psychoticism	0.648 ***	0.668 ***	0.432 **
Problematic internet use	0.353 **	0.299 *	0.197
Depression	0.589 **	0.636 *	0.436 *
Anxiety	0.429 ***	0.470 ***	0.276 *
PLEs total distress	0.423 ***	0.380 **	0.382 **

Note. PLEs: psychotic-like experiences. * *p* < 0.05, ** *p* < 0.01, *** *p* < 0.001. N = 221.

**Table 3 ijerph-19-10408-t003:** Descriptive statistics according to sex.

	Total		Male (*n* = 123)		Female (*n* = 98)		*t* (219)	Cohen’s *d*	*p*
	M	SD	M	SD	M	SD			
Total HQ-25	24.66	15.03	23.46	13.33	26.15	16.88	−1.32	0.179	0.19
Socialization	10.13	7.77	9.16	7.04	11.35	8.48	−2.09	0.283	0.04
Isolation	8.50	5.51	8.16	4.96	8.93	6.12	−1.03	0.139	0.31
Emotional support	6.02	4.03	6.14	3.81	5.88	4.31	0.48	0.065	0.63
Age	15.15	2.91	15.15	2.81	15.16	3.05	−0.04	0.006	0.97
Psychoticism	1.03	0.68	0.98	0.63	1.09	0.73	−1.19	0.162	0.23
Problematic internet use	37.54	8.97	34.96	7.74	40.78	9.39	−5.05	0.683	<0.001
Depression	0.92	0.85	0.73	0.67	1.16	0.97	−3.94	0.534	<0.001
Anxiety	0.75	0.70	0.58	0.56	0.95	0.80	−3.95	0.535	<0.001
PLEs total distress	18.51	14.03	17.16	12.87	20.19	15.27	−1.60	0.217	0.11

Note. M: mean, SD: standard deviation, PLEs: psychotic-like experiences.

**Table 4 ijerph-19-10408-t004:** Descriptive statistics according to risk for hikikomori (indicated by a HQ-25 score of 42 or above).

	Not at Risk (*n* = 194)		At Risk (*n* = 27)		*t* (188)	Cohen’s *d*	*p*
	M	SD	M	SD			
Psychoticism	0.94	0.63	1.70	0.65	−5.87	1.206	<0.001
Problematic internet use	36.99	8.20	41.44	12.78	−2.44	0.502	0.015
Depression	0.79	0.72	1.86	1.06	−6.73	1.383	<0.001
Anxiety	0.67	0.65	1.31	0.80	−4.68	0.961	<0.001
PLEs total distress	17.07	13.11	28.81	16.30	−4.23	0.868	<0.001

Note. M: mean, SD: standard deviation, PLEs: psychotic-like experiences.

**Table 5 ijerph-19-10408-t005:** Descriptive statistics for the HQ-25 according to lifetime history of hikikomori and pre-hikikomori.

	No Social Withdrawal (*n* = 168)		Hikikomori and Pre-Hikikomori (*n* = 6)	
	M	SD	M	SD
Total HQ-25	24.39	14.61	32.2	8.0
Socialization	9.91	7.60	14.0	4.9
Isolation	8.54	5.36	11.7	4.5
Emotional support	5.95	3.93	6.5	2.6

Note. M: mean, SD: standard deviation.

**Table 6 ijerph-19-10408-t006:** Completely standardized latent associations between the factors representing the independent variables of interest and distress due to PLEs as a dependent variable.

	Unstandardized Estimate (SE)	Standardized Estimate	*z*	*p*
Age	−0.026 (0.036)	−0.054	−0.726	0.468
Sex	−0.099 (0.196)	−0.035	−0.509	0.611
Psychoticism	0.575 (0.213)	0.404	2.697	0.007
Depression	−0.158 (0.257)	−0.111	−0.616	0.538
Anxiety	0.643 (0.207)	0.452	3.097	0.002
Symptoms of hikikomori	0.011 (0.173)	0.008	0.066	0.948
Problematic internet use	0.233 (0.133)	0.164	1.748	0.08
*Interaction*				
Problematic internet use × Symptoms of hikikomori	−0.123 (0.11)	−0.087	−1.125	0.261

Note. PLEs: Psychotic-like experiences, SE: standard error.

## Data Availability

The data presented in this study are available on request from the corresponding author.

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
