# Peer review of "Psychometric Properties of the Italian Version of the 25-Item Hikikomori Questionnaire for Adolescents"

_ijerph, 2022, doi:10.3390/ijerph191610408_

Round 1

Reviewer 1 Report

Thank you for the opportunity to review the manuscript entitled, “Psychometric properties of the Italian version of the 25-item hikikomori questionnaire for adolescents.” The paper is well-written and this study fills an important gap in the literature by validating the HQ-25 specifically among Italian adolescents. Strengths of the study include the thorough methods of adapting the HQ-25 and examining its psychometric properties. My major concerns are focused on the brief discussion of the potential limitations presented by the sampling strategy, as well as the reliance on cut-off scores developed for other samples. Please see below for my suggestions on ways to improve the manuscript

Introduction

1.      The sub-section titled “Symptoms of hikikomori, psychotic-like experiences and problematic internet use” might fit better if it is moved before the HQ-25 is introduced and when the authors are describing the prevalence and symptoms of hikikomori.

2.      Briefly highlighting the methodological limitations of not having a validated Italian version of the HQ-25 would strengthen the rationale for the current study.

3.      It seems that the second aim seeks to provide evidence of convergent validity. The authors should consider re-wording this aim, and the related analyses, to reference convergent validity.

Methods

1.      Please clarify when data collection occurred. Many of the items for the HQ-25, and self-reported history of hikikomori, could have been affected by public health measures due to COVID-19. If data collection occurred during COVID-19, it would be important to discuss the limitations this poses in the discussion.

2.      I was confused as to how the authors state that they do not know the number of students at each school, but then also describe the approximate proportion of students who participated. If exact recruitment rates are available, please provide them.

3.      When describing the HQ-25, it would be helpful to provide one or two sample items for each domain.

Results

1.      The Cronbach alpha and omega for the emotional support subscale are below .70 and would be more accurately described as questionable.

Discussion

1.    Given that hikikomori has been considered a culture-bound syndrome, it may be informative to include some discussion of how cultural factors may have resulted in findings that were contrary to hypotheses (e.g., cultural differences in internet use).

2.      Given that hikikomori can affect school attendance, this limitation seems much more significant and warrants additional discussion to assure readers that the sample is indeed appropriate to validate a hikikomori questionnaire.

3.      An additional limitation is that the cut-off score used in the current study is not specific to an adolescent sample. It would be important to re-establish the cut-off score using this or another sample.

Reviewer 2 Report

Thank you for the opportunity to review this article. The study examines the psychometric properties of the Italian version of the Hikikomori Questionnaire (HQ-25) in a sample of adolescents. Hikikomori, as an emerging phenomenon, deserves more attention in research and clinical practices. This calls for a psychometrically sound and locally adapted measure for screening and early intervention. The Introduction and Discussion are well-structured and easy to follow. The analyses, covering reliability, validity and factor structure, are comprehensive and sensible. I have several comments for the authors’ consideration.

Introduction:

-        The original HQ-25 has a three-factor structure (i.e. socialization, isolation and emotional support) (p. 2, lines 85-86). Please describe the conceptualizations/manifestations of these factors of hikikomori, which could let the readers understand the construct more fully.

-        Psychoticism was included as a measure in this study. However, it was not introduced and mentioned in the Introduction. Please add the description of psychoticism and the value of including it (as a personality trait) in the Introduction.

-        The examination of the interaction between problematic internet use and social withdrawal in the development of psychotic experiences needs extra theoretical formulation. The study argued problematic internet use as a stressor, which could act on social withdrawal, and contribute to the development of psychotic experiences (p. 3, lines 121-124). However, some theoretical models of psychosis consider social isolation as a stressor which triggers the development of psychotic experiences among individuals with pre-existing vulnerability (e.g. social deafferentation hypothesis). The formulation of this hypothesis needs further consideration.

Methods:

-        I wondered about the value and positioning of the self-report history of hikikomori using its proposed definition (section 2.3.2). Is it a golden standard for diagnosing hikikomori to be evaluated against the established cut-off of the HQ-25 (if this is true, what is the sensitivity and specificity of the cut-off?) Or the self-report history of hikikomori was just used for the investigation of convergent validity? Clarification of the use of the self-report history and the related analysis is needed.

Results:

-        Table 2 led to the speculation that specific factors of hikikomori are differentially associated with distinct psychopathologies. Specifically, ‘socialization’ and ‘isolation’ were more associated with psychoticism, problematic internet use, depression and anxiety; whereas all three factors seemed equally associated with psychotic experiences. The authors may consider the test of comparisons between these correlations, which could offer additional evidence for these relationships between hikikomori and psychopathologies.

Discussion:

-        The current finding in supplementary table S3 suggested that the risk of hikikomori was associated with specific psychotic experiences, in particular various delusional ideations, rather than hallucinatory experiences. This finding deserves some more discussion. This is consistent with some empirical evidence that social isolation may be more relevant to delusions (especially persecutory delusions) than other psychotic experiences. This finding highlights the need to consider the specific relationship between hikikomori with individual psychotic psychopathologies.         

Reviewer 3 Report

The literature review is very thorough and well organised, it provides a good overview of a complex topic across different dimensions. A variety of sources are used to weave a substantially validated discourse.

The article combines scientific methodology with a broad information base to produce relevant and useful research relating to the field in question. The research method is clearly described and shows good command of the main issues of the methodology, with a good take on limitations of the study. However, there is a confusion in regards to the participants involved in the HQ-25 study in Italy. The sample size includes people who are not adolescents anymore. Further details to support this statement should be provided.

Good analysis and synthesis, with a clear narrative reconnecting to the themes of the literature review and the objectives.
